# Retrospective Analysis of the Factors Affecting the Induction of Childbirth in 4350 Women from a Single Center in Warsaw, Poland

**DOI:** 10.3390/ijerph19159540

**Published:** 2022-08-03

**Authors:** Grażyna Bączek, Ewa Rzońca, Patryk Rzońca, Sylwia Rychlewicz, Margareta Budner, Agnieszka Bień

**Affiliations:** 1Department of Obstetrics and Gynecology Didactics, Faculty of Health Sciences, Medical University of Warsaw, 00-575 Warsaw, Poland; erzonca@wum.edu.pl; 2Department of Human Anatomy, Faculty of Health Sciences, Medical University of Warsaw, 02-004 Warsaw, Poland; przonca@wum.edu.pl; 3St. Sophia’s Specialist Hospital, Żelazna Medical Center, 01-004 Warsaw, Poland; s.rychlewicz@szpitalzelazna.pl; 4Eastern Center of Burns Treatment and Reconstructive Surgery, Medical University of Lublin, 20-059 Łęczna, Poland; mmbudner@wp.pl; 5Department of Plastic Surgery, Department of Jaw Orthopaedics, University Dentistry Center, University of Lublin, 20-093 Lublin, Poland; 6Clinic of Congenital Craniofacial Deformitis, 20-093 Lublin, Poland; 7Chair of Obstetrics Development, Faculty of Health Sciences, Medical University of Lublin, 20-081 Lublin, Poland; agnieszka.bien@umlub.pl

**Keywords:** delivery, induction, mother, neonate, newborn

## Abstract

Labor induction is one of the most common procedures performed during childbirth, on average in 20–30% of all pregnant women. The aim of this paper was to perform a retrospective analysis of the factors influencing the induction of childbirth. The data provide population-based evidence for Poland (Masovian Voivodeship). The electronic patient records of a hospital in Warsaw were used to create an anonymous retrospective database of all deliveries from 2015 to 2020. The study included an analysis of two groups of patients. The study group consisted of patients with labor induction—4350 cases, and the control group of patients with spontaneous contractions—20,345. The factors influencing the lower frequency of labor induction in the study group were previous cesarean section (OR = 0.73, 95% CI: 0.64–0.84, *p* < 0.05) and a higher number of deliveries (OR = 0.74, 95% CI: 0.68–0.80, *p* < 0.05). It is necessary to conduct further research about obstetric procedures used during childbirth, such as induction of childbirth, to reduce the risk of complications and improve the perinatal care of the mother and the neonate.

## 1. Introduction

Induction of Labor (IOL) was introduced into obstetric practice in the late 1700s [1]. It is an obstetric procedure that involves artificial stimulation of the mechanisms leading to childbirth before its spontaneous start [2,3]. It is one of the most frequently performed procedures during labor. In high-income countries, the percentage of newborns born after IOL is estimated at about 25%, while in low- and middle-income countries these rates are generally lower [2,4,5,6,7].

Improving the induction of labor was associated with an increasingly precise determination of the date of childbirth and at the same time a better understanding of the risks associated with prolonged pregnancy [1]. The time of delivery is an important element for pregnancy, but it should be emphasized that inducing labor is indicated in situations in which perinatal results for the mother and the newborn will be more beneficial if the pregnancy ends [2,4,8,9]. The results of the research showed that the incidence of maternal and fetal complications in the case of, for example, pregnancy complicated by intrahepatic cholestasis, hypertension, or in the case when fetal growth restriction increases with the advancement of pregnancy over 39 weeks; therefore, early completion of pregnancy in this case benefits from a previously planned labor [10,11]. When qualifying a pregnant woman for the induction of labor, the age of pregnancy should always be considered based on ultrasound performed in the first trimester, the degree of severity of the abnormalities found, parity, cervical maturation, and the occurrence of possible contraindications [1,2,4].

IOL interference in the natural process of pregnancy and childbirth in the case of medically inappropriate inductions to end pregnancy or childbirth at a specific time or to shorten the duration of labor is associated with a higher percentage of complications, such as bleeding, cesarean section, uterine hyperstimulation, or uterine rupture [7,12]. On the other hand, the results of the research indicate that the planned induction of labor in uncomplicated single pregnancy at the 39th week of pregnancy may be associated with, among others, a reduced risk of completion of childbirth by cesarean section [2.10]. The planned induction of labor also has benefits for the fetus/newborn, in the form of, among others, reduced mortality or the use of respiratory support in the newborn [4,10].

Tsakiridis et al. (2020), who reviewed guidelines on induction of labor by the American College of Obstetricians and Gynecologists’ (ACOG), the Society of Obstetricians and Gynaecologists of Canada (SOGC), the National Institute for Health and Care Excellence (NICE), and the World Health Organization (WHO), presented similarities and differences in terms of indications, contraindications, methods, and optimal time of induction. The differences in obstetric practices between and within countries result in the presence of different risk factors that increase or decrease the frequency of performing this procedure. Knowledge of these factors helps to create universal guidelines that unify the procedure in different countries [12]. In Poland, there are Recommendations of the Polish Gynecological Society regarding the induction of childbirth from 2021 [2]. Despite the WHO recommendations that the percentage of childbirth inductions should not exceed 10%, in Poland this percentage is still over 43% [13,14].

Updating the guidelines on the basis of current research is necessary to promote evidence-based medicine. Unfortunately, such research is lacking in Poland. In relation to above information, it is important to present studies on labor induction. Therefore, the authors conducted studies and performed a retrospective analysis of the factors affecting the performance of labor induction.

## 2. Materials and Methods

This was a single-center retrospective cohort study. The Strobe guideline for cohort studies was used to ensure the proper reporting of results [15]. Data were collected in a maternity hospital in an urban center with a large number of deliveries in Poland, which were then used to create an anonymous retrospective database of all the deliveries from 1 January 2015 to 31 December 2020. This dataset was generated using electronic medical records collected by medical personnel. Therefore, there is no recall bias. Additionally, the dataset was cross-checked for inconsistencies, and any detected were verified. Data on the parturient woman, the course of labor, and the condition of the newborn are recorded in a computer database by midwives during and immediately after childbirth.

The study included an analysis of two groups of patients. The study group consisted of women who had labor induction performed, and the control group consisted of patients whose contractions occurred spontaneously at the time of childbirth. The analysis of the data included information on singleton birth, in patients with induced labor at term. In the process of electronic analysis of the documentation, the following information was obtained: demographic data of women, obstetric history, previous cesarean section (information on whether the patient had at least one cesarean section), duration of pregnancy (determined on the basis of the date of the last menstrual period, confirmed by ultrasound performed before 14 + 0 Hbd), course and complications of pregnancy, data on the course of delivery, and birth data of the child (among others, a family birth, i.e., a birth in which an accompanying person participates; most often, it is the patient’s husband or partner). The inclusion criteria in the study group were as follows: childbirth between 39 and 42 weeks of pregnancy, use of induction of childbirth, and single live pregnancy. The inclusion criteria in the control group were: spontaneous initiation and completion of pregnancy between 39 and 42 weeks of pregnancy as well as single live pregnancy.

The criteria for exclusion from the study both from the study group (induction of childbirth) and from the control group (spontaneous birth) were multiple pregnancy, childbirth at less than 39 weeks of pregnancy, lack of data in the electronic documentation, stillbirth; neonates with major birth defects or abnormal karyotype were also excluded. An analysis of the documentation covering 40,007 deliveries at the analyzed time was carried out, of which, based on the adopted criteria, 24,695 cases were qualified for analysis, including the study group of 4350 and the control group of 20,345. Figure 1 shows a detailed selection of the analyzed cases.

The study received approval from the Bioethics Committee of the Medical University of Warsaw (No. AKBE/204/2021). This was a retrospective anonymized data analysis; therefore, no individual patient consent was needed.

## 3. Statistical Analysis

The collected data were subjected to statistical analysis using STATISTICA software version 13.2 (Tibco Software Inc., Palo Alto, CA, USA). The number (*n*) and percentage (%) were used to present categorical data, and the median (Me) and interquartile range (IQR) were used for continuous data. The normality of the distribution of quantitative variables was checked using the Kolmogorov–Smirnov test and the Lilliefors test. The chi-square test was used to analyze the correlation between qualitative variables, and the Mann–Whitney U test was used to compare quantitative variables. 

To measure the strength of the relationship between the dependent variable (birth induction) and the predictors, the odds ratios (ORs) were used together with a 95% confidence interval (95% CI). To analyze the factors affecting the frequency of labor induction, a multivariable regression analysis using the stepwise selection method was carried out. A single factor logistic regression analysis was conducted to identify the analyzed variables as potential risk factors for performing labor induction with oxytocin. Variables with a *p* value > 0.05 were then excluded from the multivariable model of logistic regression analysis, which were: age, place of residence, education, marital status, maternal smoking, perineal laceration, and duration of childbirth. The level of statistical significance was set at *p* < 0.05.

## 4. Results

The induction of childbirth was more often performed in younger women (31.4 vs. 31.5), single women (20.0%), with obesity (4.6%) compared to the control group (*p* < 0.05)—Table 1.

The conducted analysis showed a statistically significant relationship between the use of labor induction and the number of pregnancies, the week of pregnancy, the number of births, the occurrence of chronic diseases and diseases coexisting with pregnancy, including gestational diabetes, diabetes mellitus, pregnancy hypertension, pre-pregnancy hypertension, and pregnancy cholestasis (*p* < 0.05). There was an increased OR of labor induction in pregnant women with gestational diabetes, diabetes mellitus, pregnancy hypertension, pre-pregnancy hypertension, and pregnancy cholestasis and those with over 40 weeks of pregnancy.

In addition, a lower OR of induction in the second and subsequent pregnancy was found, as in the case of the second and subsequent childbirth—Table 2.

The analysis showed a statistically significant relationship between the performance of labor induction and a previous cesarean section, preinduction, epidural anesthesia, episiotomy family birth, type of labor, duration of labor, including the duration of the first and the second period of labor, blood loss, Apgar score at 1st minute after delivery, and the birth weight of the newborn (*p* < 0.05). However, there was no statistically significant correlation between the performance of labor induction and the duration of the third period of labor and spontaneous perineal laceration (*p* > 0.05). A lower OR of labor induction was found in the case of a previous cesarean section. However, it was found that in the group of women who underwent preinduction of labor, the chance of induction of labor increases. Induction of labor has also been shown to increase the chance of epidural anesthesia and episiotomy. There was also a higher OR of labor induction in the case of family birth. In addition, induction of labor increases the chances of operative completion of labor but reduces the chances of completion of pregnancy by cesarean section. The conducted analysis showed that in the case of induction of labor, the duration of the first period of labor was shorter, but the duration of the second period and the entire labor was longer. The abovementioned correlations were statistically significant (*p* < 0.05). Detailed data are presented in Table 3.

Table 4 presents a multivariate logistic regression analysis examining the factors affecting the performance of labor induction. The analysis shows that obesity (OR = 2.29, 95% CI: 1.86–2.81, *p* < 0.05), as well as the advanced duration of pregnancy (OR = 2.23, 95% CI: 2.11–2.34, *p* < 0.05) increased the frequency of labor induction. Moreover, diabetes (OR = 8.46, 95% CI: 2.90–24.62, *p* < 0.05), gestational diabetes (OR = 3.77, 95% CI: 3.36–4.22, *p* < 0.05), pregnancy hypertension (OR = 4.89, 95% CI: 4.01–5.96, *p* < 0.05), pre-pregnancy hypertension (OR = 3.00, 95% CI: 2.00–4.50, *p* < 0.05), and pregnancy cholestasis (OR = 9.89, 95% CI: 5.37–19.19, *p* < 0.05) were important factors influencing the more frequent use of labor induction as well as preinduction (OR = 6.02, 95% CI: 4.83–7.50, *p* < 0.05). Factors influencing the lower frequency of labor induction in the study group were previous cesarean section (OR = 0.73, 95% CI: 0.64–0.84, *p* < 0.05) and a higher number of deliveries (OR = 0.74, 95% CI: 0.68–0.80, *p* < 0.05).

## 5. Discussion

The time of delivery is an important element for pregnancy, but it should be emphasized that labor induction is indicated in situations in which perinatal results for the mother and the newborn will be more beneficial if the pregnancy ends [2,4,8,9]. The aim was to perform a retrospective analysis of the factors influencing the induction of childbirth. Our multivariate logistic regression analysis showed that obesity, as well as the advanced duration of pregnancy increased the frequency of labor induction. Moreover, diabetes, gestational diabetes, pregnancy hypertension, pre-pregnancy hypertension, and pregnancy cholestasis were important factors influencing the more frequent use of labor induction as well as preinduction. Factors influencing the lower frequency of labor induction in the study group were previous cesarean section/history of cesarean section and a higher number of deliveries.

Women with obesity are more likely to have comorbidities that may require early completion of pregnancy and induction of labor to induce contractions. In our study, it was noted that obesity in the examined women increased the frequency of labor induction. In contrast to our results, those obtained by Kim et al. showed that obesity did not affect the frequency of labor induction [4]. El-Sayed et al. reported that a higher body mass index (BMI) significantly affected the increased risk of perinatal complications in the case of induced births [16]. In turn, studies by Frolov et al. showed that obese women were more likely to have induction of labor and a prolonged second period of childbirth [17]. A detailed analysis of obesity in the context of induction and its course was presented by Ellis et al. They found that cesarean section occurred more often in women with obesity compared to women with a normal body weight during labor induction. Maternal obesity was associated with a longer duration of labor, higher doses of prostaglandins, ineffectiveness of methods causing acceleration of cervical maturation, the need to use higher doses of synthetic oxytocin, as well as a longer duration of labor after the use of oxytocin [18].

Proposing induction at 39–40 weeks is associated with a lower number of cesarean sections and a lower incidence of morbidity in the pregnant woman and the newborn compared to the wait and see attitude [19]. Studies by El-Sayed et al. indicated that the induction of childbirth at the 39th week of pregnancy was associated with a reduced risk of complications in the perinatal period [16]. In turn, Grobman and Caughey indicated that the planned induction of labor at week 39 was associated with a much lower frequency of cesarean sections and the occurrence of infections in the perinatal period [20]. Bergholt et al. proved that gestational age did not have a significant impact on the age-specific or general risk of completion of induced labor with cesarean section [21]. Subsequent studies by Grobman et al. showed that the planned induction of childbirth at the 39th week of pregnancy did not result in a higher frequency of perinatal complications than the wait and see procedure but resulted in a lower number of cases of cesarean section [22]. 

The results of our own research showed that a more advanced duration of pregnancy, over 40 weeks of pregnancy increased the frequency of induction of labor. This is in line with the principles of evidence-based medicine [2].

Batinelli et al. noted that multiparity and a high Bishop score at the beginning of prostaglandin induction were protective factors for natural birth for women over 35 years of age [23]. On the other hand, the results of our own research showed that the factor influencing the lower frequency of birth induction in the studied group was a higher number of births.

The results of our own research have shown that diseases involving pregnancy, such as diabetes, gestational diabetes, pregnancy hypertension, pre-pregnancy hypertension, and pregnancy cholestasis were important factors influencing the more frequent induction of labor. The reasons for such behavior are probably dictated by medical indications. The conducted observational and randomized studies showed that the induction of pregnancy complicated by gestational diabetes between 38 and 40 weeks of pregnancy reduces the risk of intrauterine death and cesarean delivery compared to the wait and see procedure. In cases of induction at 39 weeks of pregnancy, there was no increased risk of neonatal complications [24]. In the case of hypertension, in the absence of additional complications, delivery between 38 and 39 weeks of pregnancy compensates for the risk of maternal and neonatal complications. In cases of ineffective hypotensive therapy or restriction of intrauterine growth, early completion of pregnancy should be considered [25]. In turn, pregnancy complicated with cholestasis, despite its small impact on maternal health, is an important risk factor for the development of fetal complications, including intrauterine death, the frequency of which increases with the duration of pregnancy and the concentration of bile acids [26]. A retrospective analysis of the cohort of pregnant women with and without cholestasis indicates that childbirth at 36 weeks reduces the risk of intrauterine death and compensates for the risk of neonatal complications [27]. The studies by Delporte et al. (2019) showed that in the group of women who underwent labor induction, there was a significantly higher percentage of women with diabetes and preeclampsia [28]. Moreover, studies by Vecchiol et al. (2020) showed that among the French women examined, who used induction of labor, complications during pregnancy, in the form of diabetes or hypertension were found more often [29]. In addition, with reference to the age of women, which was an important predictor of induction of labor in our research, it should be noted that it is also the subject of interest for researchers. The phenomenon of pregnancies of women of older age is increasingly observed, which is often associated with numerous complications, such as the number of cesarean sections performed, premature births, the occurrence of diseases (gestational diabetes mellitus, gestational hypertension, and preeclampsia section) [30,31,32], all of which may justify the results we obtained.

The results of our own research showed that induction of labor was less frequently performed in women whose previous pregnancies ended in a cesarean section. The results of the systematic review and the meta-analysis carried out by Rossi and Prefumo emphasizing that the induction of labor after a cesarean section increases the risk of uterine rupture and the second cesarean section [33] are important in this respect, which may justify the results we obtained. Because vaginal delivery is successful in 73% of cases, either with spontaneous or with induced labor, a previous cesarean section should not be considered as an indication for elective cesarean section in the current pregnancy. This incidence reduces to 66% if labor is induced and increases to 74% if labor is spontaneous. However, the paucity of prospective studies and the lack of randomized clinical trials make available evidence insufficient to conclude the safety of IOL after a previous cesarean section [33,34].

The morphological and functional maturity of the cervix is the main determinant of the success of labor induction. Pre-induction cervical ripening greatly influences the outcome of induction of labor [35]. Patro-Małysza et al. (2010) conducted research on the effectiveness of preinduction of childbirth using a Foley catheter. They showed that more than four-fifths of pregnant women who underwent this obstetric procedure were induced by a drip infusion of oxytocin [36]. The results of our own research also showed that performing preinduction of labor is an important factor influencing the more frequent use of labor induction.

Our study is important for obstetric practice because it presents the characteristics of women who have received labor induction, one of the important and common obstetric procedures, as well as indicates the potential consequences for the mother and the child because of its use. However, according to our own research and also from the research of other authors, the perinatal maternal and neonatal consequences of labor induction at the 39th week of pregnancy are inconsistent [16,20,32,33].

However, as Grobman et al. suggests, one cesarean delivery may be avoided for every 28 deliveries among low-risk nulliparous women who plan to undergo elective induction of labor at 39 weeks [22].

The strong part of our study was the large sample size, covering a long period. We also did not include incomplete data in the analyses. Another advantage was the quality of the collected data. Data were collected from one institution, which reduces the risk of bias caused by differences in data collection or practices. On the other hand, research conducted in one center may be a weakness of our study, because we could not compare differences in practices and results with other providers. In our study, we also did not analyze the preparation of women for childbirth, for example in the school of childbirth, or the long-term effects on the child, such as cerebral palsy or hypoxic-ischemic encephalopathy. Our research would certainly be more valuable if we could also analyze the preferences and experiences of women giving birth.

Therefore, it is necessary to conduct further research about obstetric procedures used during childbirth, such as labor induction, to rationally use them, reduce medicalization, reduce the risk of complications, and improve the quality of perinatal care.

## 6. Conclusions

Labor induction is more often performed in older women, with an advanced duration of pregnancy, in pregnant women living in cities, as well as in the case of diseases such as diabetes, gestational diabetes, pregnancy hypertension, pre-pregnancy hypertension, pregnancy cholestasis, obesity, and the performance of preinduction. A lower incidence of induction of labor was demonstrated in the case of completion of pregnancy by cesarean section in the past and a higher number of births.

In the case of induced births, epidural anesthesia, perineal incision, and operative birth is significantly more often used. There is also a more frequent occurrence of perineal lacerations, prolongation of the second period of childbirth, and the duration of childbirth in general.

## Figures and Tables

**Figure 1 ijerph-19-09540-f001:**
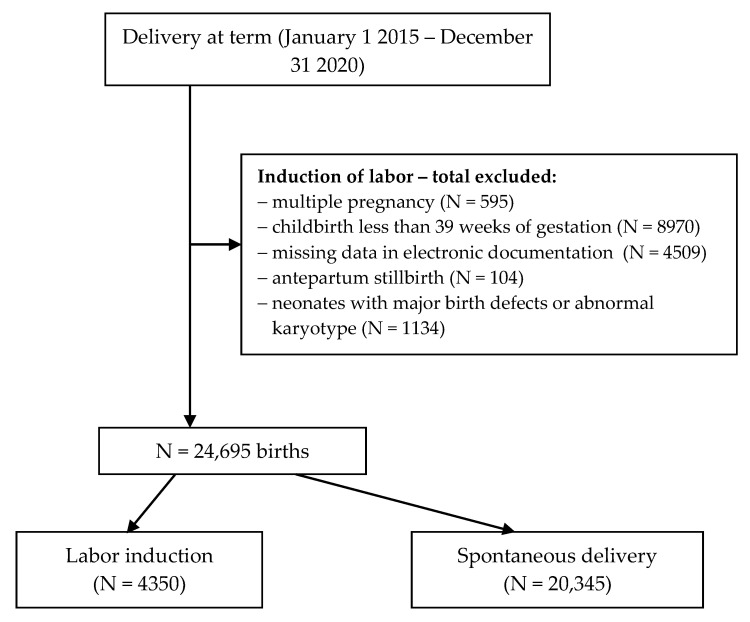
Flow diagram of exclusions and final analytic sample.

**Table 1 ijerph-19-09540-t001:** Characteristic of the populations.

Variable	Study Group Induction	Control Group No Induction	OR	95% CI	*p*-Value
Age—*n* (%)					
≤35 years	3630 (83.45)	16,854 (82.84)	1	Ref	0.334
>35 years	720 (16.55)	3491 (17.16)	0.96	0.88–1.05	
Place of residence—*n* (%)					
City	3751 (86.2)	17,657 (86.8)	1	Ref	0.325
Village	599 (13.8)	2688 (13.2)	1.05	0.95–1.15	
Education—*n* (%)					
Higher education	3778 (86.9)	17,889 (87.9)	1	Ref	0.093
Secondary	504 (11.6)	2129 (10.5)	1.12	1.01–1.24	
Primary and vocational	68 (1.6)	327 (1.6)	0.99	0.76–1.28	
Marital Status *n* (%)					
In a relationship	3479 (80.0)	16,538 (81.3)	1	Ref	0.045
Single	871 (20.0)	3807 (18.7)	1.09	1.00–1.18	
Obesity—*n* (%)					
No	4152 (95.4)	20,028 (98.4)	1	Ref	0.000
Yes	198 (4.6)	317 (1.6)	3.01	2.52–3.61	
Maternal smoking—*n* (%)					
No	4327 (99.5)	20,266 (99.6)	1	Ref	0.190
Yes	23 (0.5)	79 (0.4)	1.36	0.86–2.17	

**Table 2 ijerph-19-09540-t002:** Maternal factors occurring before induction of labor.

Variables	Study Group Induction	Control Group No Induction	OR	95% CI	*p*-Value
Gravidity—*n* (%)					
First	2276 (52.3)	7748 (38.1)	1	Ref	0.000
Second	1262 (29.0)	7643 (37.6)	0.56	0.52–0.61	
Third and subsequent	812 (18.7)	4954 (24.3)	0.56	0.51–0.61	
Week of Pregnancy—*n* (%)					
39 weeks	1272 (29.24)	10,432 (51.28)	1	Ref	0.000
40 weeks	1824 (41.93)	7773 (38.21)	1.92	1.78–2.08	
41 weeks	1233 (28.34)	2118 (10.41)	4.77	4.36–5.23	
≥42 weeks	21 (0.48)	22 (0.11)	7.83	4.29–14.28	
Parity—*n* (%)					
First	2694 (61.9)	9081 (44.6)	1	Ref	0.000
Second and subsequent	1656 (38.1)	11,264 (55.4)	0.50	0.46–0.53	
Gestational diabetes—*n* (%)					
No	3664 (84.2)	19,059 (93.7)	1	Ref	0.000
Yes	686 (15.8)	1286 (6.3)	2.78	2.51–3.06	
Diabetes mellitus—*n* (%)					
No	4338 (99.7)	20,338 (100.0)	1	Ref	0.000
Yes	12 (0.3)	7 (0.0)	8.04	3.16–20.43	
Pregnancy hypertension—*n* (%)					
No	4122 (94.8)	20,062 (98.6)	1	Ref	0.000
Yes	228 (5.2)	283 (1.4)	3.92	3.28–4.68	
Pre-Pregnancy hypertension—*n* (%)					
No	4307 (99.0)	20,262 (99.6)	1	Ref	0.000
Yes	43 (1.0)	83 (0.4)	2.44	1.68–3.53	
Pregnancy cholestasis—*n* (%)					
No	4320 (99.3)	20,324 (99.9)	1	Ref	0.000
Yes	30 (0.7)	21 (0.1)	6.72	3.84–11.75	

**Table 3 ijerph-19-09540-t003:** Selected variables and induction of childbirth.

Variables	Study Group Induction	Control Group No Induction	OR	95% CI	*p*-Value
Previous cesarean section—*n* (%)					
No	4021 (92.4)	17,017 (83.6)	1	Ref	0.000
Yes	329 (7.6)	3326 (16.4)	0.42	0.37–0.47	
Preinduction—*n* (%)					
No	4045 (93.0)	20,200 (99.3)	1	Ref	0.000
Yes	305 (7.0)	145 (0.7)	10.50	8.60–12.84	
Epidural anesthesia—*n* (%)					
No	2181 (50.1)	15,203 (74.7)	1	Ref	0.000
Yes	2169 (49.9)	5142 (25.3)	2.94	2.75–3.15	
Perineal laceration—*n* (%)					
No	3252 (74.8)	15,388 (75.6)	1	Ref	0.223
Yes	1098 (25.2)	4957 (24.4)	1.05	0.97–1.13	
Perineal incision-*n* (%)					
No	2941 (67.6)	16,158 (79.4)	1	Ref	0.000
Yes	1409 (32.4)	4187 (20.6)	1.85	1.72–1.99	
Family Childbirth—*n* (%)					
No	2863 (65.8)	15,084 (74.1)	1	Ref	0.000
Yes	1487 (34.2)	5261 (25.9)	1.49	1.39–1.60	
Type of delivery—*n* (%)					
Natural labor	3389 (77.9)	14,150 (69.6)	1	Ref	0.000
Cesarean section	810 (18.6)	5802 (28.5)	0.58	0.54–0.63	
Operative	151 (3.5)	393 (1.9)	1.60	1.33–1.94	
Duration of 1st period [min]—Me (IQR)	255.0 (175.0–345.0)	280.0 (195.0–400.0)	-	-	0.000
Duration 2nd period [min]—Me (IQR)	25.0 (15.0–40.0)	20.0 (10.0–40.0)	-	-	0.000
Duration of 3rd period [min]—Me (IQR)	10.0 (10.0–10.0)	10.0 (10.0–10.0)	-	-	0.894
Duration of delivery [min]—Me (IQR)	260.0 (145.0–380.0)	215.0 (50.0–360.0)	-	-	0.000

**Table 4 ijerph-19-09540-t004:** Multivariate logistic regression analysis of factors affecting the performance of labor induction.

Selected Predictors	*p*	Exp(B) (Odds Ratio)	95% CI	
Obesity	0.000	2.29	1.86	2.81
Week of pregnancy	0.000	2.23	2.11	2.34
* Birth	0.000	0.74	0.68	0.80
Diabetes	0.000	8.46	2.90	24.62
Gestational Diabetes	0.000	3.77	3.36	4.22
Pregnancy hypertension	0.000	4.89	4.01	5.96
Pre-Pregnancy hypertension	0.000	3.00	2.00	4.50
Pregnancy cholestasis	0.000	9.89	5.37	18.19
Previous cesarean section	0.000	0.73	0.64	0.84
Preinduction	0.000	6.02	4.83	7.50

Reference categories: * number of births—first birth.

## Data Availability

The data presented in this study are available on request from the corresponding author.

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
