# Peer review of "Retrospective Analysis of the Factors Affecting the Induction of Childbirth in 4350 Women from a Single Center in Warsaw, Poland"

_ijerph, 2022, doi:10.3390/ijerph19159540_

Round 1

Reviewer 1 Report

This study describes associations between maternal factors and odds of labor induction among term pregnancies (39+ weeks). Though the sample size is large and data seems to be high quality, overall, it is unclear what this paper adds to the existing literature on labor induction. The aim of the paper (included in the title) appears to be to evaluate maternal and neonatal outcomes following labor induction at term, but that is not what the authors do in this paper. With the exception of items in Table 3 (which I am unclear if they were analyzed as dependent variables/outcomes or independent variables/predictors), there are no maternal or neonatal outcomes presented in this paper—induction of labor appears to be the only outcome of study.  Thus, despite the stated aims, this paper presents factors that are associated with higher odds of induction of labor, which are already well-established, particularly in the term period. For example, conditions like gestational diabetes and hypertensive disorders were associated with higher odd of induction—this seems like an obvious conclusion to make, and the authors even cite literature in the discussion describing the higher risk of stillbirth and other maternal and fetal complications for women with GDM and hypertensive disorders.  Below are some general notes:

Introduction—Maybe this study provides some contribution, but the authors do not set up the need for this analysis. What is the gap in literature that is being filled? How is this study novel after large-scale clinical trials of induction of labor vs. expectant management (Grobman et al.’s ARRIVE trial)? Maybe if there was some presentation of outcomes (i.e., maternal complications of delivery or postpartum morbidities and/or neonatal outcomes), I could see the novelty, but not in its current form, presenting predictors of induction.

Materials and Methods—Missing a lot of important information on the actual predictors. I get that this was abstracted from the medical record, but the reader needs a description and definition of items like Family Childbirth and “periods” of delivery. Does that mean stages of labor? What is “duration of delivery?” Is it total length of labor?

Analysis and presentation of descriptive statistics—First, the authors use “qualitative data” and “quantitative data,” when I assumed they mean “categorical” and “continuous data.” It is all quantitative data. Second, I’m not really sure why the authors checked for normality of continuous data and then used Mann-Whitney tests (which are typically reserved for nonparametric and ordinal data) for everything. It is more customary to use t tests/ANOVA for normally-distributed continuous data, and Kruskal-Wallis tests (or other tests to compare medians) or non-normally distributed continuous data. I wonder if this is why the authors found a statistically significant difference in ages between groups, despite the means being 31.4 vs 31.5 (which is not clinically meaningful, even if it is statistically meaningful). Additionally, for variables that are skewed, they should be presented as median (IQR or range), and NOT as mean (SD).

Analysis—Author mention using a stepwise approach, but don’t mention what p-value was used for retention in the model, or any other details. What factors were excluded? In general, it’s a little hard to follow what the authors did. Do the ORs in Tables 1-3 all come from unadjusted logistic regression? I assume induction of labor is the outcome/dependent variable for all, but the presentation of Apgar scores seems a little strange (why would apgar score influence induction?) If Apgar was the outcome in that regression, it should not be in the same table as the other results where induction is the outcome. “Multivariate” should be “multivariable.” Multivariable indicates the analysis was adjusted, multivariate is a different method of analysis, despite these terms being used interchangeably in some papers.

·         Pg 3, last PP—“The OR with a CI was used to assess the strength of the association between the study group and the control group.” That’s not what an odds ratio is for. It’s to show the association between an exposure/IV and outcome/DV, by showing how different levels of the exposure change the odds of the outcome, relative to a control group. This is just an example of how it is difficult to determine what the authors did, based on this analysis section.

Results—It’s difficult to fairly evaluate these results without a more detailed Methods or Analysis section. Based on my interpretation of what was done, these are all intuitive findings, but not novel. It seems obvious that the odds of induction would increase with increasing gestational age—depending on the obstetric practice, it’s policy to induce at 41 weeks in uncomplicated pregnancies. What was the range of gestational ages in the study group? Considering that gestational age would start at 39 weeks (per inclusion criteria), it would be more informative to see the percentage in each group by week of gestation (e.g., 39, 40, 41, 42, >42 weeks, if there are women greater than 42 weeks). I don’t think providing the mean is informative, considering the tight window of gestational age in the sample.

Table 3—I think it must be mislabeled (same label as Table 2), but are these the outcomes the authors reference in the aims? It’s a little odd because, again, there’s no description of how these variables (some of which would have to occur after induction) were analyzed. On pg 5, lines 148-149 (“A lower OR  . . . was found in the case of a pregnant woman), authors appear to reference results of other studies, which is not appropriate within the Results section.

Table 4—It’s not appropriate to present R2 of logistic regression because it is not analogous to R2 in linear regression (logistic uses pseudo-likelihood methods). It is also not customary to present anything other than OR, CI, and p-value (if necessary—it seems like these p-values could be labeled in a footnote) for logistic regression.

Discussion—“The aim of our research was to analyze the relationship between induction of childbirth, selected variables and maternal and neonatal results.” The authors go on to discuss the “selected” variables, but discuss none of their results related to maternal and neonatal results in Table 3. In general, the discussion has the same issue as the introduction—it does not describe any novelty of the study or how it contributes to existing knowledge surrounding induction of labor. Some results are not supported by the data, or could be explained by other factors that the authors do not mention. For example, I’m not sure how useful it is to analyze age continuously, considering that the mean ages did not differ between the groups. Given the prior literature cited in the discussion, it seems like it may be more meaningful to analyze age categorically, looking at women over age 35 and/or over age 40.

·         Place of residence. It seems like this would depend on where you live. In some countries, it might be more difficult to access a healthcare facility in rural areas, so women may have fewer interactions with the healthcare system in general, which could indicate less likelihood of being induced.

·         Obstetric complications and higher gestational age. Induction usually indicated in the term stage of pregnancy for some complications, and even in an uncomplicated pregnancy, as the placenta ages, it loses function. Women are more likely to be induced once they've reached the 40-week mark to reduce the risk of stillbirth or other complications, so it just seems like the doctors in this hospital are following current obstetric practice.

·         Preinduction. I think it would have been surprising if the authors had found lower odds of induction in this group. Isn’t the whole purpose of preinduction to ripen the cervix for induction?

Author Response

Dear Reviewer
Thank you very much for reviewing our work, for all valuable tips and time
Best Regards

Reviewer 2 Report

The article explores an interesting topic for obstetric care. It is well presented. I only consider that the anonymity of the health institution should be maintained and only mention that the data were collected in a maternity hospital in an urban center with a large number of deliveries in Poland.

Author Response

(The authors gave the same response as above.)

Reviewer 3 Report

1.       Induction of labor at term and maternal and neonatal outcomes, in this study

correlation between the induction of labor and variables and maternal and neonatal outcomes in comparison with parturient women, in whom contractions occurred spontaneously is not enumerated in ABSTRACT. Which one is better let us Know?

2.       This study mostly elaborate about the Maternal factors responsible for  induction of labor, and the factors affecting the performance of labor induction, but no table or analysis is made about maternal and neonatal outcomes, which is your title. Explain

3.       What about the morbidity and mortality of mother and neonates in both groups, Is there any differences?

4.       Though the neonatal outcome is one of the aims, not described either in the result section or the discussion part. So, the Title may be changed to Role of maternal factors in the induction of labor or spontaneous delivery/ Factors affecting  induction of labor or spontaneous delivery … 

Author Response

(The authors gave the same response as above.)

Round 2

Reviewer 3 Report

Revised well. Concerned points addressed.

Author Response

Dear Reviewer
Thank you very much for the positive review of our work and all the comments so far, which allowed us to improve our work.
Kind regards
